

# Food additive "lauric acid" possess non-toxic profile on biochemical, haematological and histopathological studies in female Sprague Dawley (SD) rats

Hidayat Ullah Khan[1], Khurram Aamir[1], Sreenivas Patro Sisinthy[1], Narendra Babu Shivanagere Nagojappa[1] and Aditya Arya[2,3,4]

[1] School of Pharmacy, Faculty of Health and Medical Sciences, Taylor's University, Subang Jaya, Selangor, Malaysia
[2] Department of Pharmacology and Therapeutics, School of Medicine, Faculty of Health and Medical Sciences, Taylor's University, Subang Jaya, Selangor, Malaysia
[3] Department of Pharmacology and Therapeutics, Faculty of Medicine, Dentistry and Health Sciences, University of Melbourne, Melbourne, Parkville VIC, Australia
[4] Malaysian Institute of Pharmaceuticals and Nutraceuticals, (IPHARM), Bukit Gelugor Pulau, Pinang, Malaysia

Corresponding author
Aditya Arya,
aditya.arya@taylors.edu.my

## ABSTRACT

**Background**. Lauric acid (LA), a common constituent of coconut oil, is used as food additives and supplements in various formulations. Despite various potential pharmacological properties, no scientific evidence on its dose-related toxicity and safety is available till date.

**Objective**. The current study was conducted to evaluate acute oral toxicity of LA on normal rats.

**Methods**. The study was conducted in accordance with the Organization for Economic Co-operation and Development guidelines (OECD 423) with slight modifications. LA was administered orally to female Sprague Dawley (SD) rats ($n = 6$/group) at a single dose of 300 and 2,000 mg/kg body weight, respectively, while normal control received vehicle only. Animals from all the three groups were monitored for any behavioural and toxicological changes and mortality for two weeks. Food and fluid consumption, body weight was monitored on daily basis. At the end (on day 15th) of the experimental period, blood was collected for haematological and biochemical analysis. Further, all the animals were euthanized, and internal organs were harvested for histopathological investigation using four different stainings; haematoxylin and eosin, Masson trichrome, Periodic Acid Schiff and Picro Sirius Red for gross pathology through microscopical observation.

**Results**. The study results showed no LA treatment-related mortality and morbidity at two different dosages. Daily food and water consumption, body weight, relative organ weight, haematological, and biochemical analysis were observed to be normal with no severe alterations to the internal tissues.

**Conclusion**. The current finding suggests that single oral administration of LA, even up to 2,000 mg/kg body weight, did not exhibit any signs of toxicity in SD rats; thus, it was safe to be used on disease models in animals.

## INTRODUCTION

Plant-based products are considered as the primary source for food additives and ingredients in various natural products. Natural substances are added in the food to enhance its flavour, taste and appearance. Various plant additives are being used for centuries to preserve various health care products and supplements (*Amen et al., 2019*; *Arya et al., 2012a*). According to the World Health Organization (WHO) recommendation, food additives must pass through the safety screen prior to use in any products. Additives in the form of crude plant or extracts are needed to ensure processed food remains safe, and in good condition, a variety of plants extracts contain specific phytochemicals whose biological activities is not known. It is possible that some unwanted phytocompounds might lead to toxic effects or effects products preserving capacity, stability and even the nutritional value of the products (*World Health Organization, 2015*; *Yakob, Uyub & Sulaiman, 2012*). However, chemical characterization, proper dosing regimen, and insufficient knowledge of the toxicity data constraint limit the usage of additives and herbal medicine (*Deng et al., 2013*). Thus, it is the current need to evaluate the possible toxicity of the plant extract and phytocompounds before its therapeutic use.

Lauric acid (LA) is the primary (>50%) composition of coconut oil (*Sheela et al., 2016*). It is a medium-chain fatty acid (MCFA) found abundantly in the coconut oil, palm kernel oil and laurel oil, which constitutes about half of the fatty acid (*Beare-Rogers, Dieffenbacher & Holm, 2001*; *Ubgogu, Onyeagba & Chigbu, 2006*). It is also found in various kinds of foods, such as fruits, seeds, and breast milk (*Alves et al., 2017*; *Silberstein et al., 2013*; *Uday Kumar et al., 2014*). It possesses pleiotropic biological activities, including potent anti-microbial effects (*Dayrit, 2015*), beneficial effect on the cardiovascular system (CVS) by increasing high-density lipoproteins (HDL) and reducing blood pressure and heart rate in hypertensive rats (*Ahola et al., 2017*). Moreover, it also prevents testosterone-induced prostatic hyperplasia (*Babu et al., 2010*), and promotes apoptosis in breast and cancerous endometrial cells (*Lappano et al., 2017*).

Despite various pharmacological properties, scientific evidence on the toxicological and safety profile of LA is not available. This is the first study aimed at investigating the acute toxicity of LA following the Organization of Economic Cooperation and Development (OECD 423) guideline in female Sprague Dawley (SD) rats with slight modifications. In this study, we evaluated numerous parameters, like body weight, food and water intake, haematology, biochemistry, and the histopathological changes in various tissues as shown in Fig. 1.
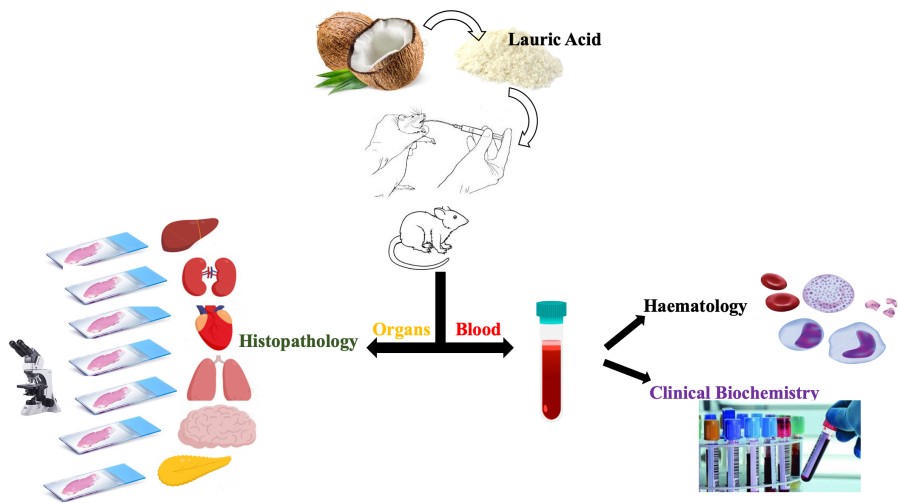

**Figure 1** Study design.

# MATERIALS & METHODS

## Ethical consideration

The acute toxicity study of LA was accomplished in accordance with the guidelines stipulated by the Institutional Animal Care and Use Committee (IACUC), Faculty of Medicine, University of Malaya, Kuala Lumpur, Malaysia with animal ethics approval no: 2018-210605/TAY/R/AA (2018295).

## Reagents

Lauric acid (MKCF8837, purity more than 98%) Sigma-Aldrich (St Louis, MO 63103 USA 314-77-5765), Tween-20 (C01-TW121-00, Systerm, UK), normal saline 0.9% (B Braun),10% neutral buffered formalin solution pH 7.4 (R&M, UK), Haematoxylin (Merck, USA) and Eosin (Systerm, UK), Masson trichrome (Abcam150686), Periodic Acid Schiff (Abcam150680) and Picro Sirius Red stain kit (Abcam 150681).

All other chemicals and solvents were of analytical grade.

## Experimental animals

In this study, eighteen (18) healthy specific pathogens-free (SPF) female SD rats, weighing 240–250 gram (g), age 8–10 weeks old were procured from the Animal Experimental Unit (AEU), Faculty of Medicine, University of Malaya. The AEU facility at the University of Malaya is certified by the Association for Assessment and Accreditation of Laboratory Animal Care International (AAALAC International). All the animals were randomly segregated in three different groups (six animals in each group) and housed in polypropylene cages for seven days to acclimatise laboratory environment prior to the experiments. A consistent environment of 12-hour light/dark cycle, temperature $22 \pm 2\,°C$, and humidity of $50 \pm 10\%$ were maintained. The animals were allowed a standard rodent pellet diet and reverse osmosis (RO) water ad libitum throughout the experimental period.

## Acute oral toxicity test

Acute toxicity study was conducted to assess whether LA produce any mortality or other adverse reactions upon oral administration of dose (300 and 2,000 mg/kg body weight), according to guideline 423 provided by the OECD for the acute toxicity class method procedure (*OECD, 2001*) with minor amendments (*OECD, 2002*). Before the experiment, all the animals were carefully weighed, marked, and grouped accordingly:

Group I ($n = 6$): Vehicle (Tween 20/normal saline)-treated

Group II ($n = 6$): Lauric acid-treated, dose 300 mg/kg body weight

Group III ($n = 6$): Lauric acid-treated, dose 2,000 mg/kg body weight.

The LA was dissolved in Tween 20 and diluted with normal saline 0.9% to get the desired dose (*Alves et al., 2017*). Animals fasted overnight before dosing, and treatment groups received freshly prepared single doses of LA suspension at 300 and 2,000 mg/kg body weight, while animals in the control group received Tween 20 or normal saline 0.9% suspension equally based on individual body weight. Each gavage volume was 10 mL/kg body weight. All efforts were made to lessen experimental animals' suffering, especially in handling and oral gavage.

## Daily observations

After oral gavage, diet was withheld except water for 2–3 h while animals were monitored individually with special attention in the first half-hour till completion of first 24 h followed by daily cage-side examination for fourteen days for any signs of toxicity including general behavioural in skin, eyes, touch, activeness, and movement changes. All the changes observed were well documented before and after the gavage.

## Body weight, food, and water consumption

The body weight of the individual animal was carefully noted before and after oral administration of the LA or vehicle, then weekly and lastly on day fifteen (euthanized day). Daily food (g/day), and water (ml/ day) intake was noted on daily basis in all the three groups for fourteen days.

## Haematology and biochemical analysis

On day fourteen, all the animals fasted overnight for 12 h, only water was allowed. All the animals were euthanized in the morning by an intraperitoneal (IP) injection of ketamine (80 mg/kg) & xylazine (7 mg/kg), and blood was collected via micro-capillary tubes from the retro-orbital plexus region into two different sets of test tubes, one containing anticoagulant, ethylene diaminotetraacetic acid (EDTA) for haematological tests including red blood cells (RBC), white blood cells (WBC), platelets count, haemoglobin (Hb), packed cell volumes (PCV), mean corpuscular volume (MCV), mean corpuscular haemoglobin (MCH), mean corpuscular haemoglobin concentration (MCHC), neutrophil, lymphocytes, monocytes, and eosinophils and the other without additives for clinical biochemistry tests like urea, creatinine, calcium, inorganic phosphate, uric acid, sodium, potassium, chloride, total cholesterol (TC), high-density lipoproteins (HDL), low-density lipoproteins (LDL), triglycerides (TG), total (HDL) ratio, albumin, globulin, albumin-globulin (A/G) ratio,
ALP, AST and ALT levels. The collected blood samples were then sent to an ISO-certified blood-testing laboratory.

## Measurement of relative organ weight

Internal organs such as adrenal glands, brain, eyes, heart, kidneys, liver, lungs, ovaries, spleen, stomach, urinary bladder, uterus, and cervix/vagina were isolated through a fine incision in the abdomen at the midline of the euthanized animals. All the harvested tissues were cleaned with phosphate buffer saline (PBS), wiped with clean blotting paper and weighed on a calibrated weighing balance. Relative organ weight (ROW) of all the animals was noted in proportion to body weight according to the following equation:

$$\text{ROW} = \frac{\text{absolute organ weight}}{\text{body weight at the time of sacrifice}} \times 100$$

## Histopathological analysis

Histopathological examination was conducted on the harvested liver, kidney, heart, lungs, brain, spleen, and pancreas in all the animals. Tissues were preserved in 10% neutral, buffered formalin solution for 48 h at 4 °C for histopathological study. Paraffin-embedded specimens of selected tissues were cut into 4–5 $\mu$m thick samples and stained with haematoxylin and eosin (H&E), Masson trichrome, Periodic Acid Schiff (PAS) and Picro Sirius Red staining as per the manufacturer protocol detailed in the kits. Later, all the stained tissues were sectioned and observed under the fluorescence compound microscope (40X) (Eclipse Ni-U, Nikon Corporation, Japan).

## Statistical analysis

All statistical results were expressed as the mean $\pm$ standard deviation (SD). The comparison between the treated and control group was performed by a one-way analysis of variance (ANOVA) using statistical software, SPSS package version 22 (IBM, SPSS Statistics, Inc., Chicago, IL, USA) to evaluate the significant differences. The ($p < 0.05$) compared to the control group was considered statistically significant.

# RESULTS

## Acute toxicological evaluation and daily observation

Animals in LA (300 and 2,000 mg/kg body weight) treated group were normal and healthy during fourteen days study period. No signs of mortality or sudden death were observed compared to the untreated group animals. No significant changes in the temperature, skin, eye colour, food and water intake were observed. Furthermore, no behavioural changes or signs of diarrhoea, coma, sedation and tremors were noted as compared to the control group Table 1. Experimental results suggest that LD$_{50}$ of LA should be higher than 2,000 mg/kg body weight.

## Effect of LA on body weight, food and water consumption

Body weight, food & water intake in the LA treated groups (300 and 2,000 mg/kg body weight) were noted to be non-significant compared to the control group. However, LA treated groups showed slight changes in the body weight, food and water consumption on day fourteen, compared to the untreated group, as shown in Tables 2 and 3. The mean

**Table 1  The behavioural and general response of animals after oral administration of LA.**

| Observation | Control | LA 300 mg/kg | LA 2,000 mg/kg |
|---|---|---|---|
| Temperature | n | n | n |
| Skin changes | no | no | no |
| Change eyes colour | no | no | no |
| General physique | n | n | n |
| Diarrhoea | no | no | no |
| Coma | no | no | no |
| Drowsiness | no | no | no |
| Breathing difficulty | no | no | no |
| Sedation | no | no | no |
| Tremor | no | no | no |
| Death | no | no | no |

Notes.
$n$, normal;  no,  not observed.

**Table 2  Effect of LA on body weight (g) assessment.**

| Week | Control | LA 300 mg/kg | LA 2,000 mg/kg | F-value | P-value |
|---|---|---|---|---|---|
| Day 0 | 244.71 ± 3.13 | 247.53 ± 2.96 | 240.63 ± 4.76 | 1.464 | 0.263 |
| Week 1% | 7.64 ± 0.46 | 7.52 ± 0.22 | 7.87 ± 0.35 | 0.149 | 0.863 |
| Week 2% | 14.81 ± 0.064 | 14.93 ± 0.68 | 15.36 ± 0.76 | 0.113 | 0.894 |

Notes.
Values are expressed as the mean standard deviation ($n = 6$ for each group, female). One-way ANOVA (*$p < 0.05$) significant value.

**Table 3  Effect of LA on food and water consumption.**

| Week | Control | LA 300 mg/kg | LA 2,000 mg/kg | F-value | P-value |
|---|---|---|---|---|---|
| **Food (g)** | | | | | |
| Week 1st | 136.35 ± 5.55 | 150.38 ± 19.65 | 133.84 ± 9.18 | 0.265 | 0.570 |
| Week 2nd | 139.50 ± 13.13 | 123.07 ± 54.79 | 142.61 ± 12.82 | | |
| **Water (ml)** | | | | | |
| Week 1st | 204.28 ± 23.17 | 211.42 ± 31.58 | 200.71 ± 67.23 | 2.609 | 0.071 |
| Week 2nd | 184.28 ± 13.36 | 172.85 ± 7.55 | 187.14 ± 14.67 | | |

Notes.
Values are expressed as the mean ± standard deviation ($n = 6$ for each group, female). One-way ANOVA (*$p < 0.05$) significant value.

body weight of the control group was noted as 244.71 ± 3.13 g, whereas treatment groups showed 247.53 ± 2.96, and 240.63 ± 4.76 g, respectively. On day fourteen, the mean body weight gained by the control group was 14.81 ± 0.064 g, while LA treated animals (300 and 2,000 mg/kg body weight) displayed non-significant gain in the body weight (14.93 ± 0.68 and 15.36 ± 0.76 g), respectively. The percentage mean in the body weight of all the treated animals was non-significant in comparison to the control group ($p > 0.05$), as shown in Table 2.

**Table 4   Effect of LA on relative organ weight (g).**

| Organ | Control | LA 300 mg/kg | LA 2,000 mg/kg | F-value | P-value |
|---|---|---|---|---|---|
| Adrenal gland (left) | $0.018 \pm 0.001$ | $0.019 \pm 0.002$ | $0.019 \pm 0.002$ | 0.638 | 0.698 |
| Adrenal gland (right) | $0.020 \pm 0.004$ | $0.018 \pm 0.003$ | $0.019 \pm 0.001$ | 0.537 | 0.595 |
| Brain | $0.643 \pm 0.032$ | $0.626 \pm 0.069$ | $0.632 \pm 0.051$ | 0.162 | 0.852 |
| Eye (left) | $0.041 \pm 0.002$ | $0.042 \pm 0.003$ | $0.044 \pm 0.003$ | 1.632 | 0.228 |
| Eye (right) | $0.043 \pm 0.001$ | $0.043 \pm 0.002$ | $0.045 \pm 0.004$ | 1.218 | 0.323 |
| Heart | $0.277 \pm 0.016$ | $0.290 \pm 0.019$ | $0.286 \pm 0.012$ | 0.960 | 0.405 |
| Kidney (left) | $0.285 \pm 0.012$ | $0.300 \pm 0.024$ | $0.280 \pm 0.025$ | 1.480 | 0.259 |
| Kidney (right) | $0.288 \pm 0.014$ | $0.297 \pm 0.020$ | $0.275 \pm 0.013$ | 2.640 | 0.104 |
| Liver | $2.981 \pm 0.254$ | $2.918 \pm 0.367$ | $2.931 \pm 0.344$ | 0.062 | 0.940 |
| Lungs | $0.476 \pm 0.038$ | $0.469 \pm 0.028$ | $0.502 \pm 0.029$ | 1.693 | 0.217 |
| Ovary (left) | $0.032 \pm 0.008$ | $0.031 \pm 0.005$ | $0.033 \pm 0.010$ | 0.062 | 0.940 |
| Ovary (right) | $0.033 \pm 0.005$ | $0.033 \pm 0.009$ | $0.033 \pm 0.006$ | 0.003 | 0.997 |
| Pancreas | $0.292 \pm 0.043$ | $0.359 \pm 0.074$ | $0.306 \pm 0.056$ | 2.117 | 0.115 |
| Spleen | $0.220 \pm 0.018$ | $0.209 \pm 0.027$ | $0.195 \pm 0.013$ | 2.284 | 0.136 |
| Stomach | $0.754 \pm 0.111$ | $0.762 \pm 0.184$ | $0.939 \pm 0.238$ | 1.890 | 0.185 |
| Urinary bladder | $0.037 \pm 0.003$ | $0.041 \pm 0.007$ | $0.037 \pm 0.004$ | 0.922 | 0.419 |
| Uterus | $0.398 \pm 0.054$ | $0.445 \pm 0.211$ | $0.258 \pm 0.315$ | 2.579 | 0.109 |
| Cervix/Vagina | $0.081 \pm 0.017$ | $0.075 \pm 0.011$ | $0.074 \pm 0.008$ | 0.445 | 0.445 |

**Notes.**

Values are expressed as the mean $\pm$ standard deviation ($n = 6$ for each group, female). One-way ANOVA ($*p < 0.05$) significant value.

Moreover, LA administration (300 and 2,000 mg/kg) showed non-significant changes ($p > 0.05$) in the food and water consumption, compared to the control group, as shown in Table 3.

## Effect of LA on relative organ weight

The relative organ weight of the harvested tissues—adrenal glands, brain, eyes, heart, kidney, liver, lungs, ovaries, spleen, stomach, urinary bladder, uterus and cervix/vagina, revealed non-significant changes post-LA treatment (300 and 2,000 mg/kg body weight) compared with the control group animals ($p > 0.05$), as shown in Table 4.

## Effect of LA on haematology and clinical biochemistry

The effect on haematological analysis is summarised in Table 5. Parameters including RBC, WBC, platelets count, Hb, PCV, MCV, MCH, MCHC, neutrophil, lymphocytes, monocytes, and eosinophils shows non-significant changes ($p > 0.05$) after oral administration of LA (300 and 2,000 mg/kg body weight) when compared with the control animals. Importantly, all the values were within the reference range, compared to the control group.

Similarly, the biochemical analysis of liver and renal function tests, including urea, creatinine, calcium, inorganic phosphate, uric acid, sodium, potassium, chloride, TC, HDL, LDL, TG, total HDL ratio, albumin, globulin, A/G ratio, ALP, AST and ALT levels for LA treatment groups were not affected and observed to be normal (non-significant ($p > 0.05$)), when compared with the control group as presented in Table 6.

**Table 5  Effect of LA on haematology.**

| Parameters | Control | LA 300 mg/kg | LA 2,000 mg/kg | F-value | P-value |
|---|---|---|---|---|---|
| RBCs ($10^{12}$/L) | 7.350 ± 0.187 | 7.616 ± 0.584 | 7.666 ± 0.216 | 1.232 | 0.320 |
| WBCs ($10^9$/L) | 3.883 ± 0.491 | 3.866 ± 1.237 | 3.533 ± 0.960 | 0.163 | 0.851 |
| Platelets ($10^9$/L) | 953.333 ± 68.677 | 865.000 ± 356.083 | 846.666 ± 154.749 | 1.091 | 0.361 |
| Hb (g/dl) | 13.716 ± 0.183 | 13.783 ± 0.523 | 13.833 ± 0.265 | 0.553 | 0.587 |
| PCV (HCT%) | 44.666 ± 1.210 | 36.833 ± 18.082 | 44.500 ± 1.378 | 0.393 | 0.682 |
| MCV fL | 52.333 ± 16.366 | 57.500 ± 2.345 | 57.333 ± 2.732 | 0.259 | 0.775 |
| MCH pg | 18.000 ± 0.894 | 17.666 ± 1.211 | 17.500 ± 0.836 | 0.377 | 0.692 |
| MCHC g/dL | 31.333 ± 0.816 | 31.500 ± 0.836 | 31.166 ± 0.752 | 0.260 | 0.775 |
| Neutrophil (%) | 24.000 ± 6.131 | 25.500 ± 7.582 | 32.500 ± 5.319 | 3.002 | 0.080 |
| Lymphocyte (%) | 64.000 ± 5.966 | 67.666 ± 7.004 | 61.833 ± 4.215 | 1.528 | 0.249 |
| Monocytes (%) | 4.333 ± 1.632 | 3.500 ± 1.760 | 3.500 ± 1.643 | 0.492 | 0.621 |
| Eosinophils (%) | 2.500 ± 0.547 | 2.333 ± 0.516 | 2.33 ± 0.516 | 0.200 | 0.821 |

**Notes.**

Values are expressed as the mean standard deviation ($n = 6$ for each group, female). One-way ANOVA ($*p < 0.05$) significant value.

**Table 6  Effect of LA on clinical biochemistry.**

| Clinical biochemistry | Control | 300 mg/kg | 2,000 mg/kg | F-value | P-value |
|---|---|---|---|---|---|
| Urea (mmol/L) | 6.633 ± 1.040 | 6.966 ± 0.977 | 8.200 ± 1.240 | 3.427 | 0.059 |
| Creatinine (Umol/L) | 36.000 ± 4.898 | 28.833 ± 13.044 | 37.000 ± 5.656 | 1.579 | 0.239 |
| Calcium (mmol/L | 2.530 ± 0.062 | 2.500 ± 0.109 | 2.403 ± 0.095 | 3.126 | 0.073 |
| Inorganic Phosphate (mmol/L) | 3.195 ± 0.350 | 3.401 ± 0.642 | 3.900 ± 0.536 | 2.871 | 0.088 |
| Uric Acid (mmol/L) | 0.211 ± 0.290 | 0.106 ± 0.048 | 0.098 ± 0.041 | 0.813 | 0.462 |
| Sodium (mmol/L) | 145.00 ± 2.097 | 145.833 ± 1.471 | 145.333 ± 2.338 | 0.263 | 0.772 |
| Potassium (mmol/L) | 5.566 ± 0.163 | 5.900 ± 0.433 | 5.616 ± 0.248 | 2.105 | 0.156 |
| Chloride(mmol/L) | 98.500 ± 1.048 | 99.000 ± 2.449 | 100.500 ± 2.073 | 1.711 | 0.214 |
| TC (mmol/L) | 2.033 ± 0.436 | 2.300 ± 0.517 | 1.966 ± 0.307 | 1.012 | 0.387 |
| HDL(mmol/L) | 0.623 ± 0.082 | 0.640 ± 0.089 | 0.590 ± 0.098 | 0.474 | 0.632 |
| LDL (mmol/L) | 0.868 ± 0.198 | 0.933 ± 0.377 | 0.733 ± 0.398 | 0.550 | 0.588 |
| TG (Umol/L) | 1.783 ± 0.231 | 1.483 ± 0.487 | 1.116 ± 0.636 | 2.879 | 0.087 |
| Total HDL Ratio | 3.383 ± 0.231 | 3.516 ± 0.444 | 3.400 ± 0.236 | 0.309 | 0.739 |
| Total protein (g/dL) | 70.833 ± 2.926 | 71.500 ± 3.507 | 66.833 ± 3.816 | 3.236 | 0.068 |
| Albumin (g/dL) | 42.166 ± 4.445 | 44.500 ± 4.888 | 43.333 ± 3.076 | 0.461 | 0.639 |
| Globulin(g/dL) | 24.333 ± 1.211 | 25.333 ± 1.211 | 23.666 ± 0.816 | 3.519 | 0.056 |
| A/G ratio | 1.800 ± 0.894 | 1.800 ± 0.109 | 1.833 ± 0.121 | 0.192 | 0.827 |
| AP (IU/L) | 53.666 ± 5.750 | 57.166 ± 11.669 | 68.33 ± 25.453 | 1.293 | 0.303 |
| SGOT /AST (IU/L) | 165.166 ± 10.703 | 200.166 ± 35.204 | 174.33 ± 18.250 | 3.315 | 0.056 |
| SGPT /ALT(IU/L) | 36.666 ± 1.966 | 36.000 ± 3.741 | 40.500 ± 19.065 | 0.278 | 0.761 |

**Notes.**

Values are expressed as the mean ± standard deviation ($n = 6$ for each group, female). One-way ANOVA ($*p < 0.05$) significant value.

## Effects of LA on Histopathology of internal organs

The histopathological analysis involves macroscopic and microscopic examination of processed and fixed tissues to visualise various components and structural alterations in the tissues (*Greaves, 2011*; *Khoo et al., 2018*). H&E staining is the most common and widely used technique to essentially identify morphological changes during diagnosis and screening of any malignancies in the tissues (*Fischer et al., 2008*). However, Masson trichrome staining remains reliable for the cytoplasm, which gives in-depth and selective detail about the connective tissue (*Tsounapi et al., 2017*). PAS staining assesses the changes in glycogen content within muscle fibre sections (*Arya et al., 2015*; *Fairchild & Fournier, 2004*). Moreover, the collagen content is analysed with the help of Picro Sirius Red staining, which provides a clear understanding of collagen accumulation in the tissues (*Lattouf et al., 2014*).

In brief, macroscopic analysis of liver sections (Fig. 2) in LA-treated rats (300 and 2,000 mg/kg body weight) displays no alterations in the colour or texture, importantly no changes to hepatocytes, hepatic sinusoids, central vein, and portal triad were observed when compared with the control group (Fig. 3). Kidney showed no structural degeneration, and colour changes in the renal cortex and tubules, without any tubular necrosis and inflammation in LA-treated animals compared to untreated animals, as displayed (Figs. 4 and 5). The photomicrographs of the heart (Figs. 6 and 7) tissue showed a normal size of myocardial fibre and filaments when compared with the control group. Lung tissues (Figs. 8 and 9) appeared to be normal, with no abnormal changes noted in colour. Moreover, microscopic observation displayed thin inter-alveolar septa, clear alveoli and alveolar sacs with the absence of inflammatory cells, including neutrophils, or thickening of alveolar septa, when compared with the untreated group. Furthermore, brain (Figs. 10 and 11) tissues showed normal morphology with the regular shape of the pyramidal neuron and uniform distribution of glial cells in LA-treated and control animals. Macroscopic texture (Figs. 12 and 13) exhibited normal and regular appearance with white and red pulp without any sign of apoptotic, necrotic lesions or megakaryocytosis compared with the control group. Microscopic appearance of the pancreas (Figs. 14 and 15) appeared with healthy acinar, clear round or oval-shaped Islets cells in cluster form, with lightly stained cytoplasm in the LA-treatment and control animals. Thus, the study results suggest that treatment with minimum and maximum doses of LA (300 & 2,000 mg/kg body weight) did not cause any tissue damage or morphological alteration at the cellular level.

## DISCUSSION

Toxicological studies of plant-based ingredients or food additives are important to ensure processed food remains safe and in good condition throughout its preservation to avoid any disease conditions. Herbs or plants additives used in dietary supplements, nutraceuticals, should be investigated for toxicity, despite their traditional claims (*Figueredo et al., 2018*; *Lulekal et al., 2013*).

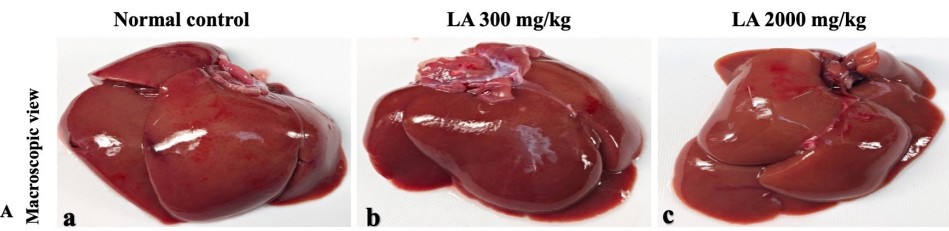

**Figure 2** **Macroscopic view of liver tissues of female SD rats in acute oral toxicity study (A) Normal control, (B) LA 300 mg/kg, (C) LA 2,000 mg/kg.** LA treated animals presenting normal reddish colour closed to the normal control group, after single oral doses of LA (300 mg/kg and 2,000 mg/kg).

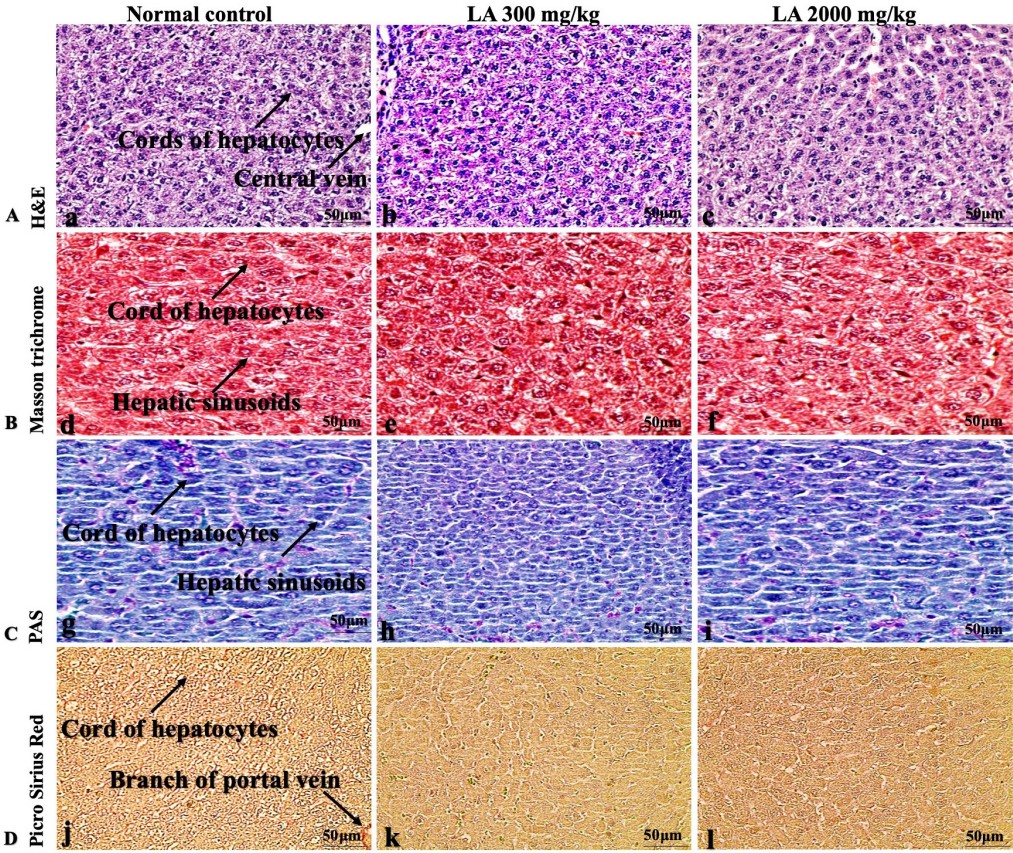

**Figure 3** **Microscopic analysis of liver tissues of female SD rats in acute oral toxicity study, presenting normal morphology after single oral doses of LA (300 mg/kg and 2,000 mg/kg).** H&E (A), Masson trichrome (B), PAS (C), and Picro Sirius Red (D). 40× original magnification.

Lauric acid revealed various pharmacological activities, but its safety profile still remains unevaluated. The current study is undertaken to screen the safety, and toxicity of LA at two different dose levels in normal rats, following OECD 423 guidelines.

No morbidity or mortality was noticed in LA-treated animals throughout fourteen days of the experimental period. Moreover, in line with previous study of *Arya et al. (2012b)*,

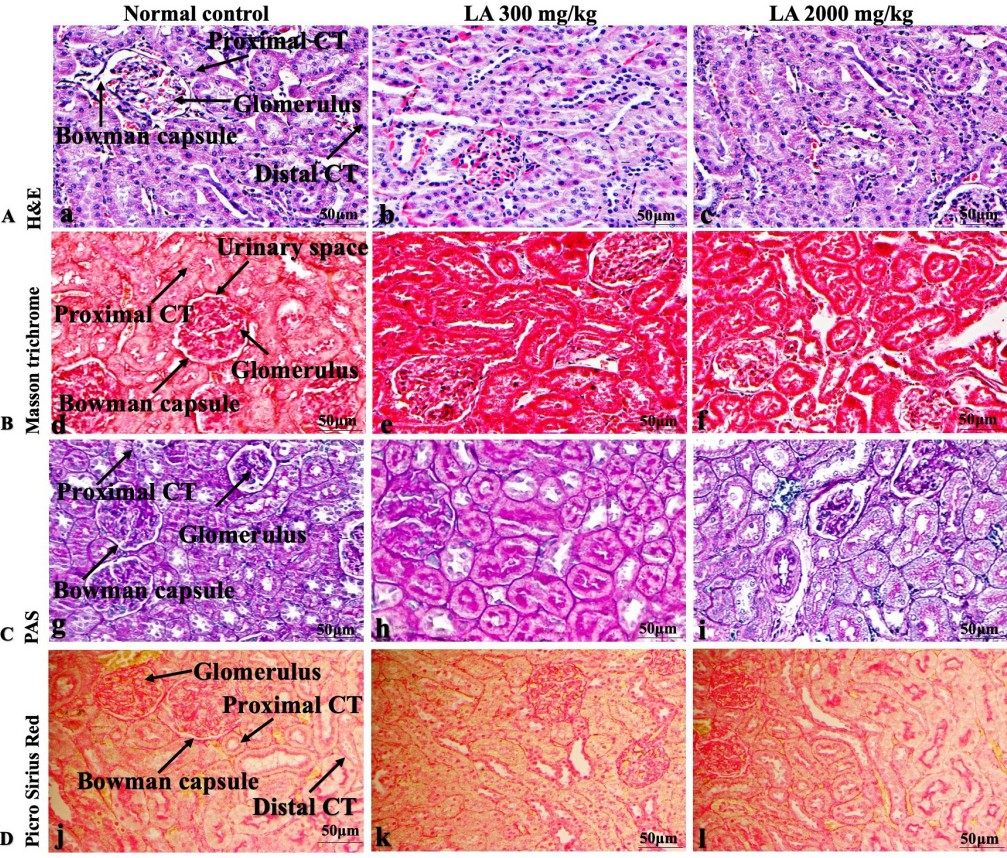

**Figure 4** Macroscopic view of kidneys of female SD rats in acute oral toxicity study, presenting normal shape and appearance after single oral doses of LA (300 mg/kg and 2,000 mg/kg) (A) Normal control, (B) LA 300 mg/kg, (C) LA 2,000 mg/kg.

**Figure 5** Microscopic analysis of kidney tissues of female SD rats in acute oral toxicity study, presenting normal morphology after single oral doses of LA (300 mg/kg and 2,000 mg/kg). H&E (A), Masson trichrome (B), PAS (C), and Picro Sirius Red (D). 40× original magnification.

no changes were observed in general behaviour of treated animals when compared with the control group, suggesting that oral ingestion of LA neither cause any alterations nor produced any internal changes in female SD rats with doses up to 2,000 mg/kg body weight.

Body weight changes are the basic parameter which indicates the health status of animals. However, abrupt variations in the body weight indicate the toxic nature of the chemicals

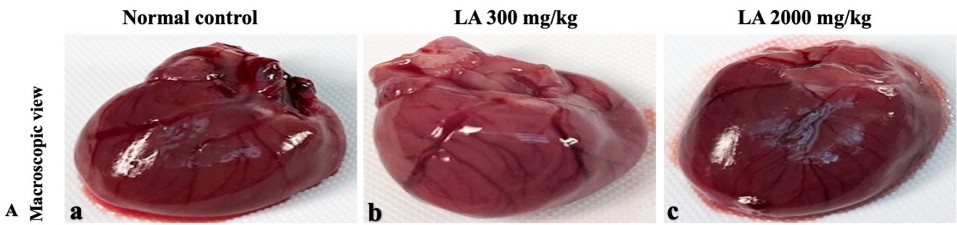

**Normal control**  **LA 300 mg/kg**  **LA 2000 mg/kg**

Macroscopic view

A  a  b  c

**Figure 6** **Macroscopic view of heart of female SD rats in acute oral toxicity study, presenting normal shape and appearance after single oral doses of LA (300 mg/kg and 2,000 mg/kg) (A) Normal control, (B) LA 300 mg/kg, (C) LA 2,000 mg/kg.**

or drugs (*Dongmo et al., 2019*). In this study, LA treatment and control group recorded a normal gain in body weight, indicating, LA does not cause any physiological alterations that affects the normal growth in the animals, which is in agreement with the previous study published by *Clemente et al. (2019)*. Similarly, organ weight variation is the fundamental and perceptive factor correlating organ to body weight index, which is important and critical in determining drug toxicity linked to treatment-related effects (*Sellers et al., 2007*). It is believed that medicinal plants in high dose could cause toxic or even lethal effects when administered in vivo (*Silveira, Bandeira & Arrais, 2008*; *Siwe et al., 2015*). Results from this study showed no difference in ROW in the treatment group when compared to control group, which indicates that LA is safer to the internal organs, supported by an earlier study published by *Porwal, Khan & Maheshwari (2017)*. Studies have proven that ingesting toxic substances could suppress appetite, thus leading to less caloric intake (*Madingou et al., 2016*). During fourteen days observation period, no significant changes were noted in water and food consumption (Table 5), which shows that LA has no effect on the appetite of the animals, which is in line with the previous findings published by *De Oliveira et al. (2020)*; *Tonholo et al. (2020)*.

The haematopoietic system is crucial in the production of blood cells that accelerate metabolic functions and enhance normal proliferation in the living organism which act and delineate as a vital tool. This system is profound and directly or indirectly susceptible to chemical-induced injury, affecting physiological and pathological status in both humans and animals (*Da Silva et al., 2018*; *Traesel et al., 2014*). In the current findings, LA administration indicate non-significant changes in the essential blood cells and related parameters RBC, WBC, platelets count, Hb PCV, Mean MCV, MCH, MCHC, Neutrophil, Lymphocyte, Monocytes and Eosinophils, suggesting non-toxic nature of LA on haematological parameters and substantiating an understanding of normal haematopoiesis, which is in line with the previous findings published by *Deyno et al. (2020)* and *Thai et al. (2020)*.

The liver is the central metabolic organ playing a key role in body metabolic functions where drug and chemical metabolism, biotransformation and detoxification occurs (*Doherty, 2016*). When hepatic cellular damage occurs due to hepatotoxic substances, the sensitive markers including transaminase serum enzymes ALT & AST start releasing from the cytosol and translocate into the bloodstream, leading to mild or severe liver

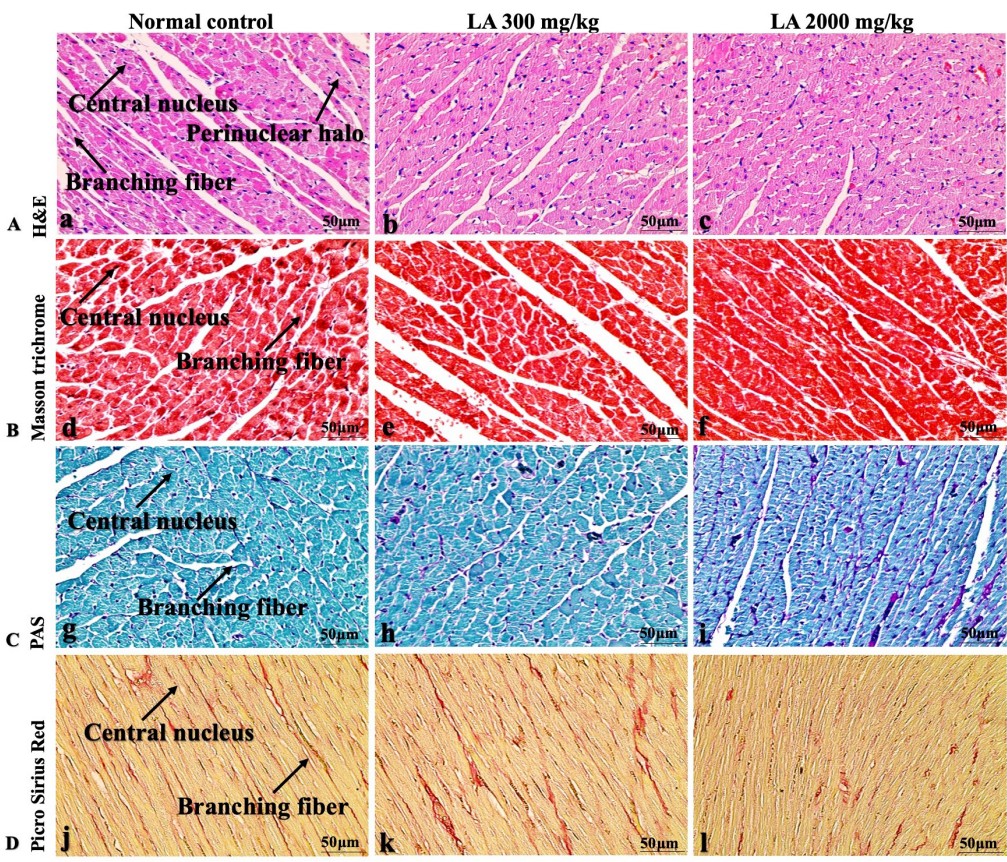

**Figure 7** **Microscopic analysis of heart tissues of female SD rats in acute oral toxicity study, presenting normal morphology after single oral doses of LA (300 mg/kg and 2,000 mg/kg).** H&E (A), Masson trichrome (B), PAS (C), and Picro Sirius Red (D). 40× original magnification.

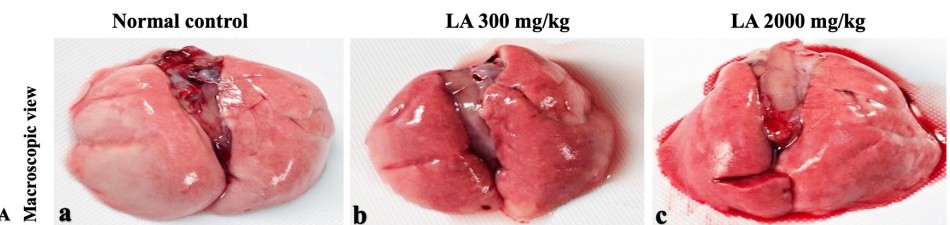

**Figure 8** **Macroscopic view of lungs of female SD rats in acute oral toxicity study, presenting normal shape and appearance after single oral doses of LA (300 mg/kg and 2,000 mg/kg) (A) Normal control, (B) LA 300 mg/kg, (C) LA 2,000 mg/kg.**

injury (*Figueredo et al., 2018*; *Hazarika et al., 2019*; *Ramaiah, 2011*). In our study, LA did not elevate these transaminase enzymes, considering non-toxic nature of LA at different dose levels. Similar findings were noted by *Moorthy, Khoo & Palanisamy (2019)*, where oral administration of the geraniin-enriched extract at 2,000 mg/kg body weight did not affect the biochemical parameters. Similarly, LA treatment did not change renal functions

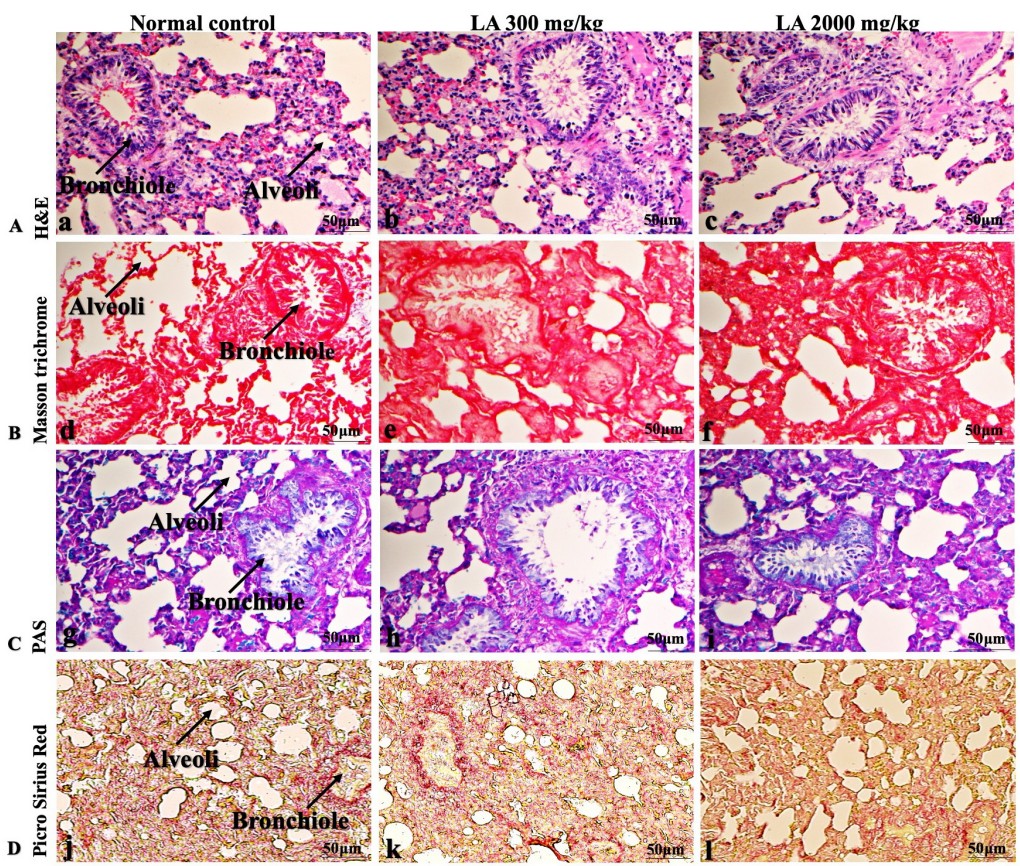

**Figure 9** Microscopic analysis of lungs tissues of female SD rats in acute oral toxicity study, presenting normal morphology after single oral doses of LA (300 mg/kg and 2,000 mg/kg). H&E (A), Masson trichrome (B), PAS (C), and Picro Sirius Red (D). 40× original magnification.

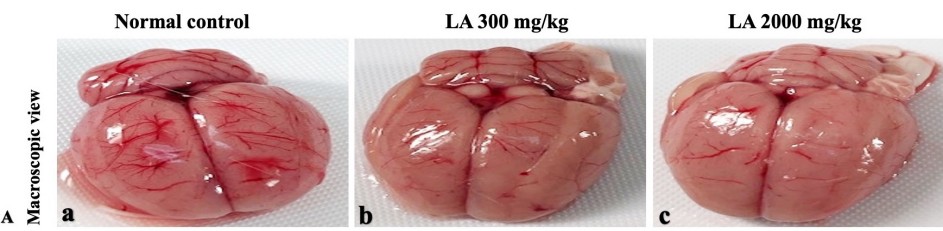

**Figure 10** Macroscopic view of brain of female SD rats in acute oral toxicity study, presenting normal shape and appearance after single oral doses of LA (300 mg/kg and 2,000 mg/kg) (A) Normal control, (B) LA 300 mg/kg, (C) LA 2,000 mg/kg.

including serum creatinine and urea levels, which are used as an essential index to estimate renal damage (*Arya et al., 2014*; *Mariappan et al., 2011*; *Meguellati et al., 2019*). The elevated level of creatinine and urea in the blood displays malfunctioning of the kidney (*Brondani et al., 2017*; *Feriani et al., 2017*; *Kifayatullah et al., 2015*). In addition, serum electrolyte levels of LA treated animals observed to be in normal ranges. Overall, biochemical parameters
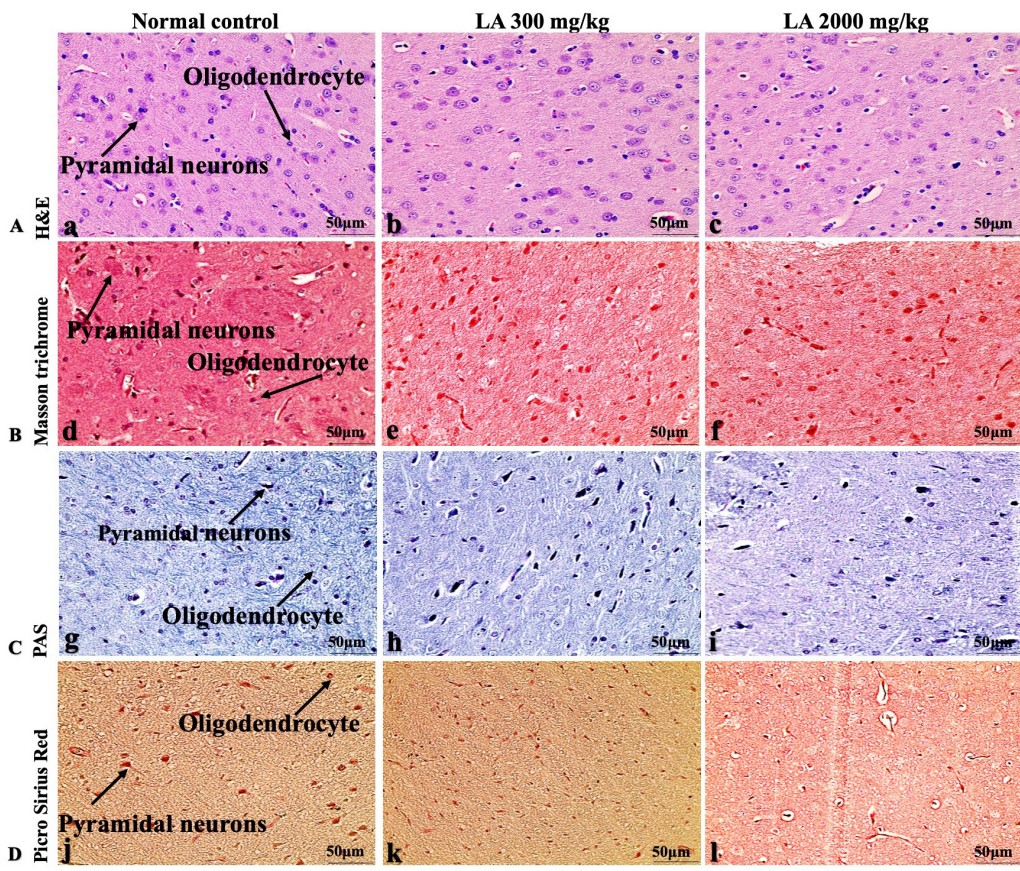

**Figure 11** **Microscopic analysis of brain tissues of female SD rats in acute oral toxicity study, presenting normal morphology after single oral doses of LA (300 mg/kg and 2,000 mg/kg).** H&E (A), Masson trichrome (B), PAS (C), and Picro Sirius Red (D). 40× original magnification.

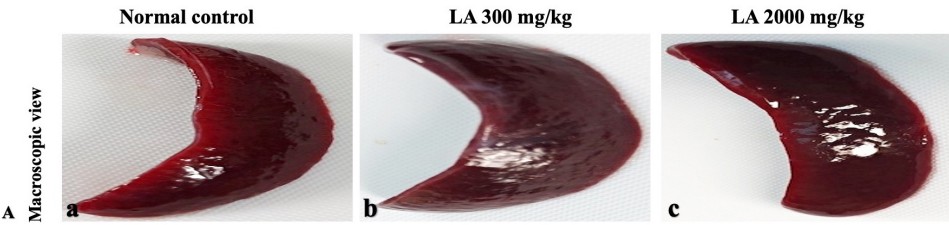

**Figure 12** **Macroscopic view of spleen of female SD rats in acute oral toxicity study, presenting normal shape and appearance after single oral doses of LA (300 mg/kg and 2,000 mg/kg) (A) Normal control, (B) LA 300 mg/kg, (C) LA 2,000 mg/kg.**

are in line with previous reports of *Bedi & Krishan (2019)* and *Thinkratok, Suwannaprapha & Srisawat (2014)*.

Histopathological investigation plays a significant role in determining tissue morphology through microscopical evaluations. The analysis of internal organs are identical and

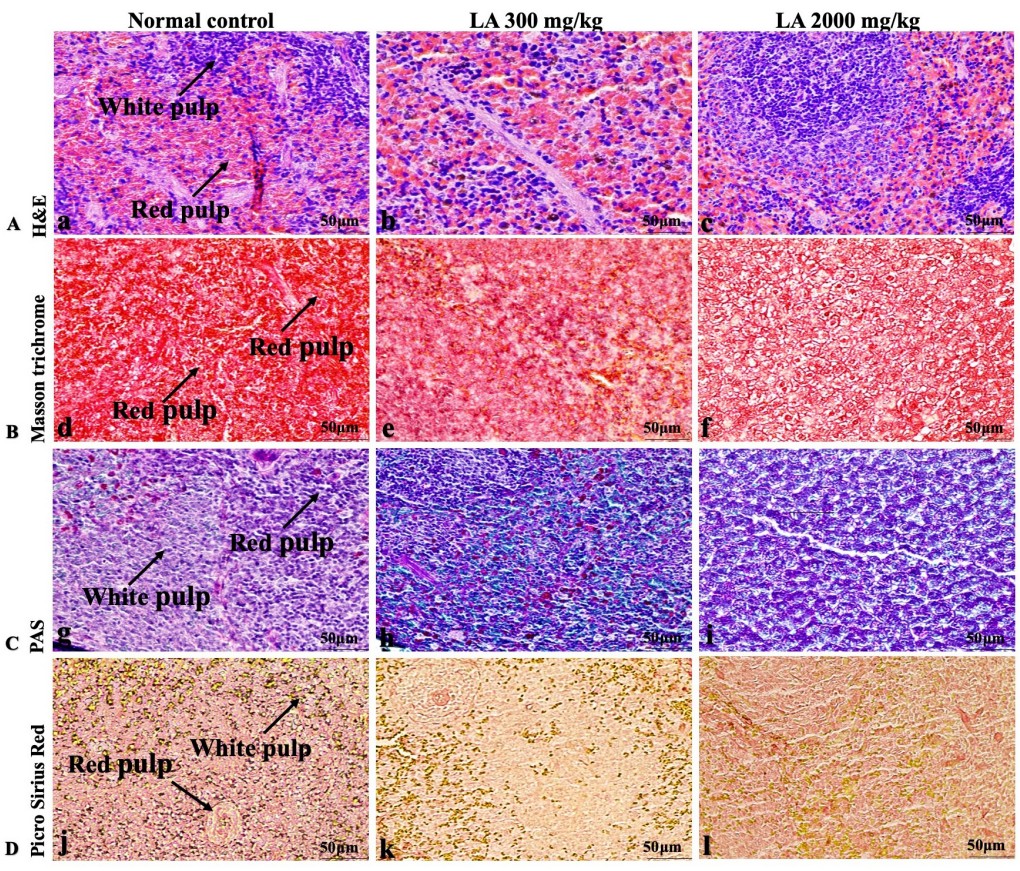

**Figure 13  Microscopic analysis of spleen tissues of female SD rats in acute oral toxicity study, presenting normal morphology after single oral doses of LA (300 mg/kg and 2,000 mg/kg).** H&E (A), Masson trichrome (B), PAS (C), and Picro Sirius Red (D). 40× original magnification.

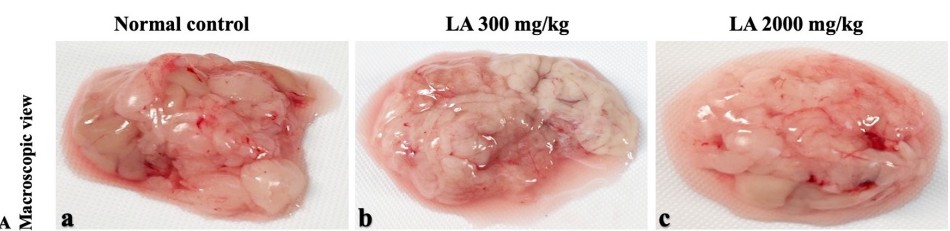

**Figure 14  Macroscopic view of pancreas of female SD rats in acute oral toxicity study, presenting normal shape and appearance after single oral doses of LA (300 mg/kg and 2,000 mg/kg) (A) Normal control, (B) LA 300 mg/kg, (C) LA 2,000 mg/kg.**

consider as the prime safety tool to determine any histological perturbations or deformities in the tissues (*Chitra, Ramaswamy & Suba, 2015*; *Prabu, Panchapakesan & Raj, 2013*). Therefore, the histopathological judgements concrete the haematological and biochemical findings and further substantiate the biological response factors (*Jaganathan et al., 2012*).

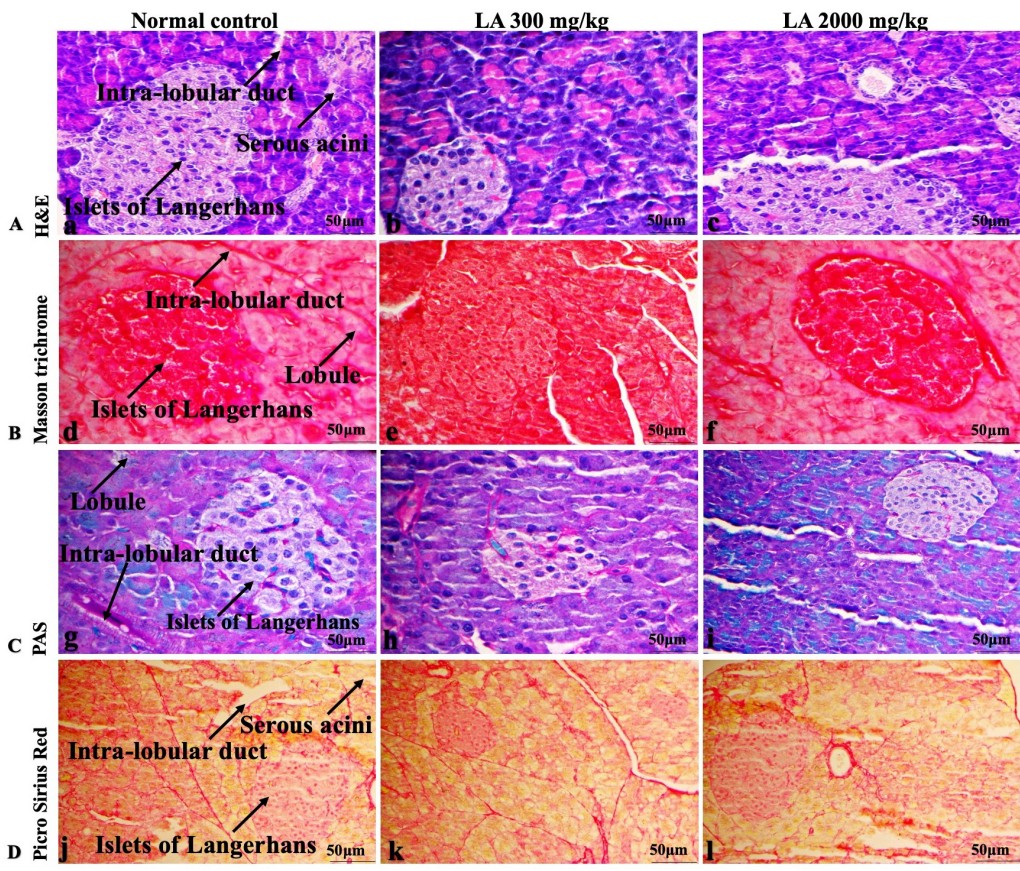

**Figure 15** **Microscopic analysis of pancreas tissues of female SD rats in acute oral toxicity study, presenting normal morphology after single oral doses of LA (300 mg/kg and 2,000 mg/kg).** H&E (A), Masson trichrome (B), PAS (C), and Picro Sirius Red (D). 40× original magnification.

Histopathological examination (macroscopic and microscopic) of internal organs including liver, kidney, heart, lungs, brain, spleen, and pancreas revealed a regular architecture, no morphological alterations were observed in LA treatment groups compared to the control group. Tissue degeneration or anomalies leading to necrosis, apoptosis or fibrosis are the basic parameters to determine toxicity in internal organs (*De Lima et al., 2017*).

The liver is the metabolic connective tissue which controls biliary and vascular traffic. Administration of LA at the minimum and maximum dose did not cause structural degeneration or accumulation in the liver tissues, displaying normal clusters of hepatocytes and parenchymal cells housed in cord-like arrangements. In this study, microscopical features of liver tissue is in agreement with the published data by *Khoo et al. (2018)* who postulated that changes to these cellular events might potentially be associated with the accumulation of exogenous chemicals and their metabolites may cause dose-dependent toxicity or damage to the liver (*Debelo et al., 2016*).

Similarly, LA administration did not alter kidney tissue, which showed organised pyramidal structure without any tubular necrosis and inflammation in the cortex

and inner portion of the medulla region. However, the toxic effect of any compound on the kidney is demonstrated by changes in the serum creatinine or urea level (*Gopi, Jacob & Mathur, 2016*). Finding from this study reveals the harmless nature of LA on renal function by establishing the outcomes from histopathological analysis, which did not notice lesion or necrotic marks upon LA administration. Likewise, LA treatment on heart tissues displayed normal architecture of myocardial fibre and filaments which is in line with *Greaves (2011)* findings, who reported no morphological changes in heart tissues upon treatment with the phytocompounds. In a similar way, lungs of LA-treated animals showed no treatment-related morphological alterations, and our findings are in agreement with *Aamir et al. (2019)* who reported that no toxicological alteration was noted upon Arjunolic acid treatment in normal female SD rats. Furthermore, brain tissues showed normal morphology upon LA treatment, with no alterations to the cell types approaching through normal characteristics based on tissue pattern recognition, including neurons, glia, and microglia cells without any modulation to perivascular region. Thus, our study is supported by previous findings by *Pravallika et al. (2019)* who reported no structural and morphological changes in brain tissues following treatment with ethanolic extract of *Centella asiatica* plant. Moreover, LA treatment did not alter pancreas and spleen tissue which shows normal architecture, as we know rodent pancreas and spleen are diffused and soft compare to human tissues, which is in line with the previously published study by *Prabu, Panchapakesan & Raj (2013)* that reveals no morphological variation in tissues of control and hydroalcoholic extract of polyherbal treatment groups.

The present study expressed that oral administration of LA did not cause any morbidity or mortality at the selected doses, and LD50 values were greater than 2,000 mg/kg body weight. Based on the current findings, LA should remain in toxicity class 5, which exhibited the lowest toxicity (LD50 > 2,000 mg/kg body weight) as per the OECD 423 guideline, under chemical labelling and classification (*OECD, 2001*).

## CONCLUSIONS

The findings of the current study showed that LA did not cause any significant changes upon oral administration up to 2,000 mg/kg body weight in female SD rats; no treatment-related signs of toxicity or mortality were noted. Moreover, histopathological examination exhibits no remarkable changes in the harvested tissues. Thus, it provides basic information on the toxicity and safety profile of LA, indicating low toxicity in the analysed parameters. This study could be helpful in the selection of dosages for future subacute or chronic disease model. However, in future, well-designed treatment strategies on pharmacological aspects should be carried out to set a clear understanding of safety and efficacy. Therefore, further validation on LA would enable to decide effective therapeutic dose prior to its long-term treatment as well as before the development of any health care products.

## ACKNOWLEDGEMENTS

We are thankful to the University of Malaya for providing animal facilities.

### Funding

The current study was funded by the Taylors University Flagship Research Grant (TUFR/2017/002/01) under the umbrella of Ageing and Quality of Life. The funders had no role in study design, data collection and analysis, decision to publish, or preparation of the manuscript.

### Grant Disclosures

The following grant information was disclosed by the authors:
Taylors University Flagship Research Grant: TUFR/2017/002/01.

### Competing Interests

The authors declare there are no competing interests.

### Author Contributions

- Hidayat Ullah Khan conceived and designed the experiments, performed the experiments, analyzed the data, prepared figures and/or tables, authored or reviewed drafts of the paper, and approved the final draft.
- Khurram Aamir performed the experiments, analyzed the data, prepared figures and/or tables, authored or reviewed drafts of the paper, and approved the final draft.
- Sreenivas Patro Sisinthy and Narendra Babu Shivanagere Nagojappa analyzed the data, authored or reviewed drafts of the paper, and approved the final draft.
- Aditya Arya conceived and designed the experiments, authored or reviewed drafts of the paper, provided instrument and chemicals for smooth operations in laboratory work, and approved the final draft.

### Field Study Permissions

The following information was supplied relating to field study approvals (i.e., approving body and any reference numbers):

The study was approved by Institutional Animal Care and Use Committee IACUC, Faculty of Medicine University of Malaya, ethics number: 2018-210605/TAY/R/AA(2018925)

### Data Deposition

Raw data is available in the Supplemental Files.

### Supplemental Information

Supplemental information for this article can be found online at http://dx.doi.org/10.7717/peerj.8805#supplemental-information.

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
