# Peer review of "Food additive “lauric acid” possess non-toxic profile on biochemical, haematological and histopathological studies in female Sprague Dawley (SD) rats"

_PeerJ, doi:10.7717/peerj.8805_

## Round 0.1 · original submission · Major Revisions

Based on the advice received, I have decided that your manuscript could be reconsidered for publication should you be prepared to incorporate major revisions suggested by the reviewers.

I would ask you to respond at all to the points raised by the reviewers.

·

Basic reporting

This study contains the acute toxicological evaluation of lauric acid, a commom food additive. The authors performed the experiments following the recommendations of OECD (423 - acute oral toxicity) but underestimated the possibilities of investigation. A few parameters were assessed and they are not enough to affirm the absence of toxicity.

The authors affirmed that no data concerning the potential toxicological properties of LA are available; however, I found the following studies about this substance: “Final Report on the Safety Assessment of Oleic Acid, Laurie Acid, Palmitic Acid, Myristic Acid, and Stearic Acid” and Safety assessment of widely used fermented virgin coconut oil (Cocos nucifera) in Malaysia: Chronic toxicity studies and SAR analysis of the active components” (DOI:10.1016/j.yrtph.2016.10.004).

Avoid using the word “sacrifice” (line 163) because it is not correct. This term is not technical. Replace it by euthanasia.

The whole manuscript lacks a high standard quality of the English language. The text contains several no-technical terms and sentences as well as syntaxis errors. Thus, I am strongly recommending a detailed revision of English grammar.

Experimental design

The authors carried out the experiments in accordance with the OECD to acute oral toxicity. However, I identified some important issues that must be taken into account:

It is not correct to affirm that no toxicological effects were caused by the administration of LA based on relative organs weigh, some plasma biochemical markers of function and histological evaluations. The oxidative status of the tissues, mainly liver and kidney, could be assessed in order to discard any impairment due to metabolization and elimination of LA. Thus, it would be interesting recognize such limitations.

The reagents used to the histological staining should be informed in the correct section (drugs and reagents) instead of the histopathological analysis. Please, correct it and provide additional information about the dyes used.

The age of the animals should be mentioned.

How the authors selected the doses used in the study? What were the scientific criteria followed?

Locomotor and coordination behaviors would provide reliable evidence concerning the occurrence of toxic effects caused by substances administration. Did the authors perform any behavioral evaluation with the animals?

In the section 2.7 – the authors should mention the biochemical parameters that were determined. Such information was provided in the result section. Please, modify it to make easier to the reader.

In the section 3.2, the following sentence “…on the day fourteenth of LA administration,…” is giving an incorrect idea of repeated administration. Please, revise it.

In the section 3.5, the authors did not adequately describe the results of histological evaluation. In the current form, the morphological characteristics of the tissues were barely mentioned. Instead of describing the tissue features in the discussion (line 309 to 343), the authors must add those data herein, in the correct section.

Validity of the findings

Concerning the validity of the findings, in the discussion, the real scientific contribution and impact of the data are not clear. The authors failed in demonstrating the importance of their study to the field of toxicology. This section is a more extensive format of the results. The authors poorly linked their results with scientific literature, which must be improved using additional studies to support the conclusions. Additionally, a paragraph of potential limitations would fit well due to the lack of more accurate techniques.

Additional comments

The current study contemplates the investigation of acute oral toxicity of lauric acid (LA), a commom food additive. Despite the adequate guideline (OECD 423), the authors collected a small piece of evidence regarding the possible toxic effects caused by LA. The manuscript needs to be doubly-checked to improve the English language and the discussion as well. I suggest the authors a detailed revision of the document taking into consideration the comments made by me.

Reviewer 2 ·

Basic reporting

no comment

Experimental design

no comment

Validity of the findings

no comment

Additional comments

In their manuscript, Khan and colleagues report the toxic effects lauric acid, a common constituent of coconut oil and it is used as food additives. Identifying toxic effects of lauric acid is important, because it is a product used in the food and has various potential pharmacological properties. The experiments seem to be well performed and the results are sound, but some points need to be addressed by the authors, main the histological analyses.

Abstract
The sentence repeat twice: “(SD) rats”;
The authors need taking care the abbreviations, for example: B.W.;
Keywords:
Please remove OECD 423 of keywords section;
Introduction
The introduction section is very long, I suggest the authors reduce.
Some points the English need be revised.
Methods
Why the authors chose the doses of 300 and 2000 mg/kg?
What the reason to use female?
In the in vivo experiments the correct is using “dose” not “concentration”.
I think that 15 days after the treatment can occurs restoration of hematological and biochemistry parameters. What the authors think about this?
Please explain the use four different histological techniques.
The figures need show the same things in the control group and treated groups.
Results
Please correct this “…as shown in (table xx)”
The authors need include the F values in the results section.
What is gm?
The first sentence of “Effects of LA on Histopathology of internal organs” is about material and methods.
The authors need correct all the images, sometimes the arrow is not in the correct place. This observation is the main point that authors need to correct.

Discussion
A discussion about toxic natural products could be interesting;

Figures
Please check all figures, mainly the arrows.
Please provide the same points of the organs in all groups;

Reviewer 3 ·

Basic reporting

The manuscript of Khan et al. is well written, however some questions need to be clarified.
Why did the authors decide to conduct toxicology testing only on females?
What is the impact of using only females for this research? Is it that many studies today show how males and females respond differently to pharmacological and toxicological treatments?

Experimental design

What werelauric acid concentrations based on?
To Daily observations, how the authors evaluated clinical signs of toxicity?based on what?

Validity of the findings

Are very well described, with no suggestions to expose.

Additional comments

The manuscript is very clear and with consistent results, after some changes I recommend this article for publication.

---

## Round 0.2 · accepted · Accept

I have received a re-review on your manuscript. Based on the advice received, I have decided that your manuscript could be accepted for publication in its present form.

·

Basic reporting

The authors took into consideration all the suggestions and queries raised by me to perform the document revision. The current version of the manuscript is now suitable for publication in the journal.

Experimental design

Overall, all my concerns were solved and properly answered. I have no additional comments.

Validity of the findings

The authors attended to my suggestion. The discussion section is now clearer and more informative in the current version of the document.

Additional comments

Dear authors,
The revision made in your manuscript considerably improved its quality and readability. Thus, I am suggesting the acceptance of the document for publication in the PeerJ.